# A Warp-Knitted Light-Emitting Fabric-Based Device for In Vitro Photodynamic Therapy: Description, Characterization, and Application on Human Cancer Cell Lines

**DOI:** 10.3390/cancers13164109

**Published:** 2021-08-15

**Authors:** Elise Thécua, Laurine Ziane, Guillaume Paul Grolez, Alexandre Fagart, Abhishek Kumar, Bertrand Leroux, Gregory Baert, Pascal Deleporte, Maximilien Vermandel, Anne-Sophie Vignion-Dewalle, Nadira Delhem, Serge Mordon

**Affiliations:** 1U1189-ONCO-THAI-Laser Assisted Therapy and Immunotherapies for Oncology, CHU Lille, Inserm, Univ. Lille, F-59000 Lille, France; laurine.ziane@inserm.fr (L.Z.); guillaume.grolez@inserm.fr (G.P.G.); alexandre1.fagart@chru-lille.fr (A.F.); abhishek.kumar@ibl.cnrs.fr (A.K.); bertrand.leroux@inserm.fr (B.L.); gregory.baert@inserm.fr (G.B.); pascal.deleporte@inserm.fr (P.D.); maximilien.vermandel@univ-lille.fr (M.V.); nadira.delhem@ibl.cnrs.fr (N.D.); serge.mordon@inserm.fr (S.M.); 2MDB TEXINOV, F-38110 Saint-Didier de la Tour, France; 3Department of Nuclear Medicine, CHU Lille, F-59000 Lille, France

**Keywords:** photodynamic therapy, light-emitting fabric, in vitro device, optics, optical fibers, laser, LED

## Abstract

**Simple Summary:**

While photodynamic therapy appears to be a promising approach to treating cancers, the complexity of its parameters prevents wide acceptance. Accurate light dose measurement is one of the keys to photodynamic effect assessment, but it remains challenging when comparing different technologies. This work provides a complete demonstration of the technical performance of a homemade optical device, based on knitted light-emitting fabrics, called CELL-LEF. Thermal and optical distributions and related safeties are investigated. The results are discussed in relation to the requirements of photodynamic therapy. The usability of CELL-LEF is investigated on human cancer cell lines as a proof of concept. This study highlights that new light-emitting fabric-based technologies can be relevant light sources for in vitro photodynamic therapy studies of tomorrow.

**Abstract:**

Photodynamic therapy (PDT) appears to be a promising strategy in biomedical applications. However, the complexity of its parameters prevents wide acceptance. This work presents and characterizes a novel optical device based on knitted light-emitting fabrics and dedicated to in vitro PDT involving low irradiance over a long illumination period. Technical characterization of this device, called CELL-LEF, is performed. A cytotoxic study of 5-ALA-mediated PDT on human cancer cell lines is provided as a proof of concept. The target of delivering an irradiance of 1 mW/cm^2^ over 750 cm^2^ is achieved (mean: 0.99 mW/cm^2^; standard deviation: 0.13 mW/cm^2^). The device can maintain a stable temperature with the mean thermal distribution of 35.1 °C (min: 30.7 °C; max: 38.4 °C). In vitro outcomes show that 5-ALA PDT using CELL-LEF consistently and effectively induced a decrease in tumor cell viability: Almost all the HepG2 cells died after 80 min of illumination, while less than 60% of U87 cell viability remained. CELL-LEF is suitable for in vitro PDT involving low irradiance over a long illumination period.

## 1. Introduction

Over the last decades, photodynamic therapy (PDT) has appeared to be a promising strategy in biomedical applications, such as oncology and the inactivation of pathogens. PDT relies on the interaction between the oxygen naturally present in biological tissues and a photosensitive molecule (PS) that selectively concentrates in tumor cells. This photochemical reaction, triggered by the delivery of a specific light [1], leads to the creation of cytotoxic molecules, including singlet oxygen, and induces selective cell death.

Regarding the light features, the therapeutic effect and safety of PDT depend on several features. First, the wavelength must correspond to the PS activation spectrum, while promoting deep penetration into biological tissues. Second, the light dose must be sufficient to produce a therapeutic effect. Last, the illumination scheme (exposure time, irradiance, and fractionation mode) [2] must maximize the treatment efficacy, while limiting toxicity [3,4].

In that respect, a strong effort has been made to design proper light sources for PDT. Laser systems are widely used to perform PDT [5,6,7], as their narrow spectral width and spatial light distribution lead to proper light dosimetry [8]. Because of the need to couple the source to an optical fiber, the illumination area results in a spotlight of a few cm^2^ [9,10], which challenges using laser systems on large surfaces, such as multi-well plates or large body parts. Because of cost, LED panels have overtaken discharge lamps, whose wavelength is achieved by using cut-off filters [11,12,13,14], and remain the simplest way to conduct PDT, making them one of the most used light sources in dermatological practice [15]. However, the broad spectral width and spatial light distribution of LEDs often lead to inaccurate light dosimetry [16], and the thermal effects from energy conversion efficiency remain an issue when contact is intended [17]. The emergence of new applications in the field of medical photobiology has led to the development of relevant diffusing applicators for body cavities [18,19,20,21]. Unfortunately, their use is restricted to the anatomical locations they are designed for, and none of them constitutes a generic light source solution for PDT. A new approach consisting of using optical fiber-based light-emitting fabrics (LEFs) has been investigated to optimize light delivery, while fitting a wide range of body areas [22,23,24]. Inserting optical fibers into a fabric structure with a specific pattern, inducing macrobending, promotes light leakage through the fabric surface [25]. Because the fabric is usually made of plastic optical fibers, the admissible power of LEF technology is generally limited, promoting irradiance of a few milliwatts per square centimeter (mW/cm^2^) [26].

The use of LEF is in line with the recent trend of changing the light delivery for clinical PDT [27]. Originally, the light dose was overestimated (above 25 J/cm^2^) to ensure a sufficient cytotoxic effect and quickly achieved this with irradiance over 75 mW/cm^2^ [28,29], regardless of the illumination technology used. The use of lower levels of irradiance prevents the limitation of the photochemical reaction by an early complete consumption of oxygen and supports limited toxicity without reducing the cure rates [30,31,32]. PDT at lower irradiance levels, therefore, emerges as a way to deliver treatment with improved tolerability and an equal efficacy both for preclinical studies [33,34], and the management of patients [23,35]. Going a step further by reducing the dose of light results in similar effects to a conventional dose [7,10,34], but also allows using more compact light sources [31,36,37], and reduced treatment durations.

Thus, despite multiple preclinical and clinical demonstrations that PDT is an effective and minimally invasive strategy in oncology [38,39], it suffers from the complexity of its parameters—PS used, lighting devices, or treatment regimens—preventing its widespread acceptance [40]. Whether preclinical studies involve a cell or animal model, no illumination device has yet been established as the gold standard. Furthermore, as spatial distribution of light plays a key role in the reproducibility of the results, an unsuitable light source, as well as uncontrolled thermal emissions during illumination, can lead to inconsistent PDT effects [41]. To address this question, we present a technical description and full characterization of an optical device based on knitted light-emitting fabrics and dedicated to in vitro PDT studies. Furthermore, we provide a proof of concept of the usability of this device, called CELL-LEF, through a cytotoxic study of 5-ALA-mediated PDT on two human cancer cell lines.

## 2. Materials and Methods

### 2.1. Discussion of Illumination Parameters

#### 2.1.1. Low-Irradiance Relevance for In Vitro 5-ALA PDT

Fluence rate has been shown to play a critical role in PDT through controlling tumor oxygen depletion. Indeed, the higher the fluence rate, the higher is the oxygen depletion that occurs in the tumor tissue during Photofrin-PDT [42]. Similar findings were also observed with HPPH (2-[1-hexyloxyethyl]-2-devinyl pyropheophorbide-a)-PDT on murine colon carcinoma [43].

PDT outcomes may be reduced by high oxygen depletion during the therapy. Thus, the assumption that a low fluence rate, in favor of low oxygen depletion, is associated with better PDT outcomes seems reasonable. This is consistent with the literature: HPPH-PDT with protocols that involved a fluence rate as low as 14 mW/cm^2^ provided more effective tumor destruction in murine models than protocols with fluence rates as high as 112 mW/cm^2^ [43,44].

Furthermore, in vitro 5-ALA PDT with irradiance as low as 0.116 mW/cm^2^ for a total light dose of 10 J/cm^2^ was associated with significantly greater apoptosis and lower necrosis of cells than the same light dose delivered at 21 mW/cm^2^ [45]. For three given light doses of 1.5 J/cm^2^, 3 J/cm^2^, and 6 J/cm^2^, 5-ALA PDT of glioblastoma spheroids with a fluence rate as ultra-low as 17–70 μW/cm^2^ was significantly more effective than that with fluence rates ranging from 0.42 to 1.70 mW/cm^2^ [10]. Similarly, 5-ALA PDT with ultra-low fluence rates ranging from 5.8 μW/cm^2^ to 2.8 mW/cm^2^ with 12 h of illumination at 634 nm led to complete extinction of cell viability, provided that the fluence rate was higher than 10 J/cm^2^ [37].

In clinical applications, it is essential to find the right balance between a fluence rate low enough to avoid excessive local oxygen depletion and high enough to reach biological tissue in depth [46]. This is not a problem for in vitro PDT, as cells in suspension are easily reached by light. In vitro PDT can, therefore, be achieved at low irradiances without light penetration limitation issues.

Another approach to preventing oxygen depletion involves “on–off” light treatment, as oxygen reperfusion is promoted in tumor tissues during light-free periods [33,47]. However, in addition to addressing oxygen limitations, low-irradiance PDT appears to support progressive PS activation. This is particularly true for PpIX, whose levels remain high up to 16 h in U87 cells in vitro [45]. Thus, provided that the ALA concentration is sufficient and the light delivery is low, this suggests that the synthesis and accumulation of PpIX can exceed the rate of photodegradation of PpIX. This last assumption is also supported by clinical outcomes from dermatology, for which low-irradiance PDT has been shown to promote continuous activation of PpIX [48,49].

However, efficient low-irradiance PDT requires an illumination time longer than that used in a typical in vitro PDT study to deliver the light dose.

#### 2.1.2. Temperature-Related Issues of In Vitro 5-ALA PDT Using Low Irradiance

##### Preventing Cells from Reaching Room Temperature

Efficient low-irradiance PDT is provided through extended illumination times. While cells of in vivo human models are naturally maintained at a physiological temperature of 37 °C, cells in multi-well plates removed from the incubator during illumination are not. As some cellular mechanisms are inhibited in hypothermia conditions (20–21 °C) [50], the first objective of CELL-LEF is to maintain a constant temperature optimally close to the physiological one. Indeed, avoiding temperature-related bias in cell viability during the long illumination time of the PDT treatment has been mentioned elsewhere [10,37].

##### Temperature Dependence of the Effects of PDT

Furthermore, it has been shown that hyperthermia treatment, with a temperature between 37 and 45 °C, does not damage cells, but can enhance the effect of in vitro hematoporphyrin-derivative PDT compared to PDT delivered at room temperature [51]. Similar results were observed by Yang et al., who reported a temperature dependence of the outcomes of the in vitro 5-ALA PDT antitumor effect [52]: The death rate of the PDT-treated cells did not fluctuate significantly between 29 and 38 °C, but increased when temperatures increased beyond 38 °C and decreased at 20 °C. These results were confirmed in vivo in the study of Henderson et al., where the association of Photofrin-PDT and heat treatment at 44 °C resulted in significantly higher mouse tumor control than Photofrin-PDT and heat alone.

Based on these studies, performing in vitro 5-ALA PDT at room temperature can lead to a significantly reduced cell death rate compared to rates obtained at physiological temperature.

### 2.2. Device Description

#### 2.2.1. Light-Emitting Fabric-Based Technologies

An optical fiber (OF) is a dielectric waveguide, mostly of cylindrical shape, initially designed to guide optical radiation with minimal losses from one end to the other [53]. Several methods have been developed to promote controlled light leakage through optical fibers. Multiple bendings can be kept in place, while embedded in a stable structure, such as a textile structure [25], including bendings [54,55], and/or cladding alterations [56,57,58].

Among the various textile structures that incorporate OFs, woven and knit-based LEFs have been widely described [59]. Woven fabrics consist of two interconnected orthogonal thread systems of warp and weft yarns. Thus, OFs can easily be integrated as a warp or weft yarn [60,61,62,63] or as an embroidery pattern [64,65,66], providing an emission perpendicular to the plane of the fabric on both sides of the surface, as shown in Figure 1. Knitted fabrics are made up of intertwined loops. The loops in the knitting process can be formed in rows in the production direction (warp knitting) or orthogonally to the production direction (weft knitting) [67]. Unlike weaving, knitted loops provide excessive tension that prevents the optical fiber from integrating directly as a warp or weft yarn without breaking. OFs are mainly laid as partial weft in knitted structures in straight lines or in sinusoidal patterns [68], providing an emission parallel to the plane of the fabric, as shown in Figure 2. Regardless of the textile structure, poly(methyl methacrylate) (PMMA) optical fibers are usually employed to limit the risk of breakage during the manufacturing process [59]. Optimization of light distribution remains a challenge and relies on the perfect control of bending angles and radii [62].

#### 2.2.2. Structure of CELL-LEF

CELL-LEF was home-designed to deliver homogenous light over a sufficiently large surface area to illuminate four 96-well plates simultaneously (Figure 3). The optical part of CELL-LEF was composed of two warp-knitted LEF samples (FluxMedicare^®^, MDB TEXINOV, France) with a light part, hereafter referred to as the active part, of approximately 750 cm^2^ (21 cm width and 36 cm length). The knitted fabric samples were sewn together on a white reflective fabric and inserted between two sheets of rigid, transparent plastic. These plastic plates held the two knitted LEF samples in place and provided protection from liquids and splashes, particularly during the disinfection of the device. A template was placed on the top of the device to indicate the location of the four multi-well plates (resulting from four operational areas, each approximately 110 cm^2^). The 2200 plastic optical fibers contained in the two knitted LEF samples were bundled into a metal connector, allowing CELL-LEF to be coupled to any laser source by means of two beam expanders.

CELL-LEF must maintain a constant temperature during long periods of illumination, optimally close to physiological temperature. Embedded in CELL-LEF is a heating system composed of a heating cable (originally for fragile aquarium plants and fishes) and a thermocouple controller with a digital temperature display. The heating system was set to achieve a surface temperature of around 37 °C. The temperature was continuously measured by the thermocouple, which, depending on the value, allowed temperature control by keeping the heating on (below 37 °C) or by stopping it (above 37 °C). As a result, the temperature fluctuated around the target value of 37 °C, preventing cells from hyperthermia and from reaching room temperature. A detailed view of CELL-LEF is given in Figure 4.

#### 2.2.3. Light Supply of CELL-LEF

Like all-optical fiber based-LEF technologies, knitted LEFs are not optical radiation generators. In fact, knitted LEFs redistribute the light they receive from one or more light sources. The characteristics of the optical radiation from the LEF surface are, therefore, intrinsically linked to those of the primary light source(s). Thus, the irradiance of CELL-LEF can be easily modulated according to the illumination protocol required for the in vitro studies conducted [69,70].

Similarly, the wavelength can be adjusted by appropriate selection of the primary light source, as the spectral distribution should match the absorption spectrum of the photosensitizer used in the experiments. PMMA optical fibers have a transmission window in the visible range [71], and bending-induced losses are not affected by wavelength in this spectral range [72]. Thus, provided that the light source exhibits a spectral emission contained in the visible spectrum (380–780 nm), CELL-LEF is suitable for multiple approved PS.

For all the experiments described below, CELL-LEF was lighted by a 635 nm diode laser (OncoThAI, Lille, France) to correspond to the 5-ALA PpIX activation spectrum. Moreover, for a target irradiance of 1 mW/cm^2^, the light source was set to deliver 2.6 W to CELL-LEF. This target irradiance was in line with the trend toward using low doses after long illumination with low irradiance described in Section 2.1.1.

### 2.3. Optical Characterization of Device

#### 2.3.1. Irradiance Level at the CELL-LEF Surface

Irradiance distribution at the CELL-LEF surface was controlled with a photodiode sensor (PD300, Ophir Photonics, Israel) and a power meter (Starbright, Ophir photonics, Israel). A positioning template was successively placed on each operational area to report eight uniformly distributed irradiance values. This was repeated three times to achieve a total of 96 irradiance values (3 × 8 values/operational area × 4 operational areas). From the data collected on the four active areas, the min to max percent difference was deduced according to Equation (1).
(1)$$\mathrm{Min}\;\mathrm{to}\;\max\;\mathrm{difference}\;(\%)=\frac{Irradiance_{max}-Irradiance_{min}}{Irradiance_{min}}\times 100$$

#### 2.3.2. Irradiance Distribution in 96-Well Plates

Irradiance at each end of the four 96-well plates—corresponding to 14 mm from the CELL-LEF surface—was measured with a photodiode sensor (PD300R, Ophir Photonics, Israel) and a power meter (Starbright, Ophir photonics, Israel). The mean irradiance value of the four 96-well plates was deduced.

#### 2.3.3. Spectral Distribution

IEC 60601-2-75:2017 specifies particular requirements for the basic safety and essential performance of PDT devices. As these devices often consist of separate light sources associated with beam transmission systems, the standard advises defining the central wavelength and spectrum width before and after connection to the beam transmission systems. In addition, if a PDT device involves a laser, 90% of the power emission must be within the spectral range of ± 3 nm around the target wavelength of the PS. This condition must be met for 95% of the illumination time.

According to the standard guidelines, the impact on the spatial distribution of the connection of CELL-LEF to the 635 nm laser (OncoThAI, Lille, France) was studied. The spectral distributions of the 635 nm laser alone and in combination with CELL-LEF were collected using an isotropic probe (IP85, Medlight, Switzerland) and a spectrophotometer (USB2000+ Vis-NIR, Ocean Optics, USA) connected to the OceanView 2.0 software (Ocean Optics, USA).

In addition, the stability of the spectral distribution of CELL-LEF was studied by recording spectra every 25 min during 100 min of illumination.

#### 2.3.4. Accessible Emission and User Safety

Prior to use, the optical safety of CELL-LEF needed to be determined. International standard IEC 62,471 specifies exposure limits to evaluate the photobiological safety of light sources in operating conditions. According to the standard, irradiance was measured at a standardized distance of 10 cm from CELL-LEF connected to the 635 nm laser source using a photodiode sensor (PD300RM, Ophir Photonics, Israel) and a power meter (Starbright, Ophir photonics, Israel). The irradiance (at 10 cm) and spectral distribution collected in Section 2.3.3 were used to determine the accessible emission from CELL-LEF, compared to the exposure limits defined for each risk at an illumination time of 100 min.

### 2.4. Thermal Characterization of Device

The ability of CELL-LEF to maintain a constant temperature close to physiological temperature was assessed using a thermal camera (Ti125, Fluke Corporation, USA). Thermal capture of the temperature distribution on the active area of CELL-LEF was performed, and mean, minimum, and maximum temperature values were recorded every 25 min during 100 min of illumination.

### 2.5. In Vitro Cytotoxic PDT Study Using CELL-LEF

A cytotoxic study of 5-ALA-mediated PDT on cancer cells was performed as a proof of concept of CELL-LEF. As a positive control for this experiment, we used a hepatocellular carcinoma cell line (HepG2) known to be highly sensitive to 5-ALA-mediated PDT [73,74,75,76]. We also used a human glioblastoma cell line (U87), for which studies are inconclusive, due to the unknown optimal parameters for high-efficacy PDT [77]. The use of this layer cell line is justified by two recent clinical studies, in which the addition of combined 5-ALA FGR and PDT to the standard of care provided a safe method of local tumor control for glioblastoma patients [78,79].

#### 2.5.1. Cell Culture and Photosensitization

The human hepatocarcinoma HepG2 cell line and human glioblastoma U87 cell line (ATCC) were both cultured in MEM medium supplemented with 10% heat-inactivated fetal calf serum (Gibco, Thermo Fisher Scientific, USA) and 100 µg/mL streptomycin and penicillin (Gibco, Thermo Fisher Scientific, USA). Cells were maintained in an incubator at 37 °C, 5% CO_2_, and 95% humidity. After 24 h, the medium was replaced with a new one containing 5-ALA (Sigma-Aldrich, USA). After 4 h, the medium was changed and replaced with the initial medium after washing with PBS (Gibco, Thermo Fisher Scientific, Waltham, MA, USA). Five × 10^6^ cells were cultured in 96-well plates (Corning, Amsterdam, The Netherlands) and subjected to lighting treatments.

#### 2.5.2. Photodynamic Therapy Protocols

##### Concentration and Light Dose

HepG2 and U87 cells photosensitized with 5-ALA (0, 0.2, 0.6, or 1 mM) received illumination using CELL-LEF (635 nm, 1 mW/cm^2^). Illumination time ranged from 5 to 40 min, leading to respective light doses from 0.3 to 2.4 J/cm^2^. Four groups of cells were used: Nontreated cells (NT), cells treated with 5-ALA, but without illumination (5-ALA), cells treated with illumination, but without 5-ALA (Light), and cells treated with both 5-ALA and illumination (PDT).

##### Fractionation

HepG2 and U87 cells photosensitized with 5-ALA (0.6 mM) received illumination using CELL-LEF (635 nm, 1 mW/cm^2^) for 80 min (light dose of 4.8 J/cm^2^) in 1, 2, 3, or 4 fractions with a pause time of 2 min 30 s between fractions. Four groups of cells were used: Nontreated cells (NT), cells treated with 5-ALA, but without illumination (5-ALA), cells treated with illumination, but without 5-ALA (Light), and cells treated with both 5-ALA and illumination (PDT).

#### 2.5.3. Viability Assays

HepG2 and U87 cells were assessed 24 h posttreatment (NT, 5-ALA, Light, and PDT) by a viability assay based on ATP measurement by bioluminescence (CellTiter-Glo^®^, Promega, Madison, WI, USA). A total of 5000 cells of HepG2 or U87 were seeded per well in a 96-well white-walled and clear-bottomed plate (Corning, Amsterdam, The Netherlands) and received 100 μL/well of the CellTiter-Glo mix at room temperature for 10 min, while protected from light. The bioluminescence was then read using a luminometer with MicroWIN software v4.41 (Centro LB960, Berthold Technologies, Bad Wildbad, Germany).

#### 2.5.4. Statistical Analysis

Results are expressed as the mean and standard deviation of triplicates of at least three independent experiments. The effect of ALA concentration, illumination time, and interaction on cell viability normalized on NT was assessed using a linear regression model. The effect of the interaction between the number of illumination fractions and the presence of 5-ALA was assessed using an ANCOVA model. Significance levels were set to 0.05.

All the statistical analyses were performed using the XLSTAT software (Addinsoft, Paris, France) under Excel (Microsoft Corporation, Redmond, WA, USA).

## 3. Results

### 3.1. Optical Characterization of Device

#### 3.1.1. Irradiance Level at the CELL-LEF Surface

CELL-LEF showed surface irradiance values between 0.78 and 1.23 mW/cm^2^ (mean: 0.99 mW/cm^2^; standard deviation: 0.13 mW/cm^2^). The min to max difference was 57.6%. The irradiance values measured at the eight control points of each of the four operational areas (Figure 3) are presented as box plots in Figure 5. CELL-LEF achieved the objective of delivering an irradiance of 1 mW/cm^2^ over an area large enough to illuminate four multi-well plates simultaneously.

#### 3.1.2. Irradiance Distribution in 96-Well Plates

The homogeneity of the light distribution over the operational areas is one of the parameters that can significantly affect PDT outcomes between wells, and therefore, needed to be controlled. Deviation from mean irradiance at each well end of the four 96-well plates is plotted as a mesh diagram per plate in Figure 6. CELL-LEF exhibited a mean deviation from mean irradiance of 17.6% (min: 11.3%; max: 22.5%).

#### 3.1.3. Spectral Distribution

The spectral distributions of the 635 nm diode laser (OncoThAI, France) alone and in combination with CELL-LEF were assessed according to the IEC 60601-2-75:2017 standard guidelines and are represented in Figure 7. The central wavelength of the diode laser alone was 636.08 nm. After combination with CELL-LEF, the mean central wavelength during the illumination time was 635.72 nm (min: 635.68 nm; max: 635.74 nm; standard deviation: 0.02 nm). Similarly, the full width at half maximum (FWHM) of the diode laser alone was 5.33 nm. After combination with CELL-LEF, the mean FWHM during the illumination time was 5.22 nm (min: 5.20 nm; max: 5.23 nm; standard deviation: 0.01 nm).

Combining the 635 nm diode laser with CELL-LEF contributed to a very slight decrease in the spectral width. This observation was consistent with the mode coupling theory that was described for the multimode optical fibers contained in knitted LEF samples [80]. The details are beyond the scope of this paper, and the theory of mode coupling that occurs in optical fibers from lower to higher propagation modes has been widely described in the literature [81,82]. This assumption is even more relevant as only the higher order propagation modes are most likely to be lost when the fiber is bent [83].

Nevertheless, 90% of the irradiance delivered by CELL-LEF was associated with the 635 nm diode laser ranges from 633 to 639 nm, which corresponded to ±3 nm around the 635 nm target wavelength related to PpIX. Furthermore, this was achieved for more than 95% of the illumination period as required by IEC 60601-2-75:2017.

#### 3.1.4. Accessible Emission and User Safety

CELL-LEF spectral radiances $L_{B\;}$, $L_{R}$, $L_{IR}$ and risk-weighted irradiances $E_{EFF}$, $E_{UV-A}$, $E_{IR}$ are detailed in Table 1. Since emissions were under the exposure limits defined by the IEC 62,471 standard, CELL-LEF was classified in the exempt risk group for retinal blue light hazard, retinal thermal hazard, and thermal hazard and could be used without protective equipment.

### 3.2. Thermal Characterization of Device

CELL-LEF managed to reach the defined temperature range after 17 min and maintained a mean surface temperature of 35.1 °C during 100 min of illumination. The minimum and maximum mean surface temperatures of 34.5 and 35.7 °C, respectively, measured during this illumination time, emphasized the temperature stability over time of CELL-LEF. Moreover, the overall spatial temperature distribution ranged from 31.2 to 38.5 °C.

Results of temperature monitoring and thermal distribution over the active area are represented in Figure 8. Monitoring the temperature for more than 100 min of illumination was beyond the scope of this study. However, as the heating system was designed to provide heat continuously, it is likely that a longer illumination time would lead to similar outcomes.

### 3.3. In Vitro Cytotoxic PDT Study Using CELL-LEF

#### 3.3.1. Concentration and Light Dose

Results of the HepG2 (human hepatocarcinoma) and U87 (human glioblastoma) cell line viability are presented in Figure 9 and Figure 10 as the mean of triplicate wells of three independent experiments and expressed in percentage, according to the NT control (100%). These results showed that, within the limits of the experiment, the longer the illumination time (i.e., the higher the light dose) and the higher the concentration of 5-ALA (0.2 to 1 mM), the greater the decrease in tumor cell viability, confirming the phototoxicity of 5-ALA PDT. The superior efficacy of PDT was found with the HepG2 cell line in comparison to the U87 cell line.

Linear regression showed that 5-ALA concentration, illumination time, and their interaction explained the variability of the cell viability (*p* < 0.0001) to a significant extent. For the HepG2 cell line, the illumination time and the 5-ALA concentration had no significant effect on the cell viability, whereas the interaction between the illumination time and the 5-ALA concentration had a significant negative effect (*p* < 0.0001; 95% CI 0.027 to –0.018). For the U87 cell line, the illumination time (*p* < 0.0001; 95% CI 0.004 to –0.002), the 5-ALA concentration (*p* = 0.0004; 95% CI 0.152 to –0.044), and their interaction had a significant negative effect (*p* < 0.0001; 95% CI 0.008 to –0.003).

Based on the linear regression results, PDT with 80 min illumination time after 4 h incubation in a medium containing a 5-ALA concentration of 0.6 mM should lead to a decrease in cell viability of 100% ± 2.1% for HepG2 and 56.6% ± 1.2% for U87.

#### 3.3.2. Fractionation

Results of cell viability are presented as a box plot in Figure 11 and expressed in percentage, according to the NT control (100%). These results showed that 5-ALA PDT very effectively decreased HepG2 viability, as almost all the tumor cells died after 80 min of illumination, whatever the number of fractions. On another note, the same 5-ALA PDT protocols applied on U87 cells were not as effective at inducing the same decrease in viability, since about 58.7 ± 5.1% of cell viability remained. These outcomes clearly showed that the HepG2 cell line was very sensitive to 5-ALA PDT, whereas, as expected, U87 was less responsive.

For both cell lines, ANCOVA demonstrated that the interaction between the number of illumination fractions and the presence/absence of 5-ALA explained to a significant extent the variability of the cell viability (*p* < 0.0001). In the absence of 5-ALA, cells subjected to light showed a slight increase in viability, which was significant for HepG2 (*p* < 0.0001), but not significant for U87 (*p* = 0.140). In the presence of 5-ALA, the number of illumination fractions had a significant negative effect on cell viability, whatever the cell lines (*p* < 0.0001).

This difference in cell viability between the two cancer cell lines was predicted by the linear regression model from the study of the impact of illumination time and 5-ALA concentration. In fact, this model forecast a decrease in cell viability of 100 ± 2.1% for HepG2 and 56.6 ± 1.2% for U87 with 80 min of illumination and a concentration of 5-ALA at 0.6 mM.

## 4. Discussion

In this work, a novel optical device based on warp-knitted light-emitting fabrics was presented. This device was designed for in vitro PDT involving low irradiance over a long period of illumination. This illumination protocol strategy was consistent with the recent trend of changing the mode of light delivery for clinical PDT [27]. Technical characterization of this device was performed. A cytotoxic study of 5-ALA-mediated PDT on human cancer cell lines was provided as a proof of concept.

CELL-LEF managed to deliver 0.99 mW/cm^2^ (min.: 0.78 mW/cm^2^; max.: 1.23 mW/cm^2^) for at least 100 min over 750 cm^2^, allowing simultaneous illumination of four multi-well plates. In our study, 90% of the irradiance delivered by CELL-LEF ranged from 633 to 639 nm and corresponded to ±3 nm around the 635 nm target wavelength related to 5-ALA.

Compared to commercial illumination devices dedicated to PDT preclinical studies, the target irradiance of 1 mW/cm^2^ of CELL-LEF can easily be delivered by LED systems (LEDBox, BioLambda, Brazil) [84]. However, illumination can only be performed on one multi-well plate at a time. The biomedical automatic illuminator (ML8500, Modulight, Finland) [85] has the advantage of successively illuminating each well of one multi-well plate with specific illumination parameters (wavelength, irradiance, and time). Nevertheless, such sequential illumination compromises the possibility of performing low-irradiance PDT because of the long illumination times. Indeed, this would result in excessively long experimental times and in unacceptable differences in the drug–light interval between wells in the same plate.

In comparison with homemade illumination devices, LEF technologies have not so far been able to achieve such optical performances. Indeed, although embroidered [65] and woven [22,25,60,66] LEFs showed significantly better light distribution, operational areas (Figure 3) are up to 10 times smaller than that of CELL-LEF. Other homemade systems incorporating diffusing optical fibers have been investigated. Their flexibility is undoubtedly suitable for clinical use, but such systems have, so far, shown insufficient optical performance for preclinical use in vitro. Indeed, one light blanket system exhibited good light distribution (mean: 7.4 mW/cm^2^/W; standard deviation: 1.1 mW/cm^2^/W), but over an active area of only ~9 × 9 cm^2^ [86]. Recently, an optical surface applicator was developed that integrates optical fibers into a flexible mesh applicator, but it exhibits an active area of only ~3 × 3.5 cm^2^ with a minimum-to-maximum difference of 170% [87]. Furthermore, LED systems can provide illumination to one multi-well plate, with red light distribution exhibiting a minimum-to-maximum difference of up to 100% for high irradiances, although this variation tends to decrease at lower irradiances [88]. QLEDs can also be reliable and low-cost systems to light multi-well plates [89], but they become difficult to implement when illuminating complete plates because the active area is below 1 cm^2^.

The optical connection is a critical point in the implementation of LEF technology. To properly light an LEF, the laser beam must reach the bundle of optical fibers. When considering the Gaussian distribution typically observed in a laser beam, the more optical fibers there are, the more difficult it is to optimize this connection. Indeed, one can easily imagine the difficulties in providing equal light to the more than 2200 optical fibers included in CELL-LEF. However, it should be noted that the homogeneity of light distribution can be increased by optimizing the connection with a collimating lens.

In vitro 5-ALA-mediated PDT is known to rely on a temperature-dependent mechanism [90]. Indeed, some studies have shown that performing in vitro 5-ALA PDT at room temperature (around 20 °C) leads to a significantly reduced cell death rate compared to rates obtained in the temperature range of 29 to 38 °C [52]. CELL-LEF maintained a constant mean temperature very close to the physiological temperature. On the one hand, this avoided additional cell death, due to hypothermia resulting from insufficient room temperature. On the other hand, this allowed us to carry out the 5-ALA PDT in the most efficient conditions. Therefore, using the CELL-LEF heating mechanism remains relevant in long-duration illuminations.

In our study, we first investigated the 5-ALA-mediated PDT effect on HepG2 and U87 cell lines with different values of 5-ALA concentration and light dose. Here, the superior efficacy of PDT was found on the HepG2 cell line in comparison to the U87 cell line. This difference agreed with the literature, since, unlike the U87 cell line, HepG2 is known to be very sensitive to 5-ALA-mediated PDT. Indeed, high photocytotoxicity toward HepG2 is expected with a light dose above 2 J/cm^2^ delivered with low irradiance [76]. In contrast, low doses and low irradiances, such as those we used, were described as resulting in poor death rates in human glioma cells (ACBT) [10]. Moreover, this latter study also suggested that the better efficacy of 5-ALA PDT could be achieved by decreasing the irradiance (to as low as 17 µW/cm^2^) and increasing the time of illumination (up to 24 h).

In a second step, we studied the relevance of using fractional illumination on HepG2 and U87 cell lines treated with low irradiance 5-ALA PDT for 80 min. Our results showed that 5-ALA PDT very effectively decreased HepG2 viability, as almost all the tumor cells died, whatever the number of fractions. Although fractional illumination significantly improved the photocytotoxicity of 5-ALA PDT to U87, the effect remained insufficient for complete cell death. This agreed with the literature because fractional illumination is associated with better outcomes of 5-ALA-mediated PDT [47] primarily because it prevents O_2_ depletion [91]. However, since fractional illumination aims to improve tissue reoxygenation, the low effect was expected as the PDT was already mediated here by the low irradiance delivered by CELL-LEF [92]. In addition to increasing illumination time, as previously mentioned in this discussion, both optimal incubation time and concentration of 5-ALA for U87 should be studied. As they have for HepG2 cells [76], these adjustments should optimize the efficacy of 5-ALA PDT on the U87 cell line. Another strategy, first reported for dermatology, is to heat the cells (up to 41 °C) prior to PDT treatment to enhance the production of PpIX from 5-ALA [93], and thus, improve cell death results [52,90]. Alternatively, several studies have suggested that a metronomic approach to 5-ALA PDT [10,34,94], with several separate, but closely spaced PDT sessions, gives excellent outcomes of phototoxicity to glioma cells.

However, efficient low-irradiance PDT is provided through extended illumination times, which may be difficult to transfer to intraoperative clinical applications. PDT delivered with a metronomic scheme with a small and portable light source can address this treatment time limitation [34,37,45,95]. This strategy is based on both light and PS delivery for several days. The portability of the devices dedicated to metronomic PDT would be achieved at the cost of an extremely low irradiance, which can be limited to a few microwatts per square centimeter (µW/cm^2^). On the other hand, prolonged illumination times can be easily implemented in dermatology. As with Daylight-PDT, which involves using daylight as a light source, prolonged illumination at low irradiance favors effective and painless PDT [35,96]. In addition, it also opens the prospect of using miniaturized and/or portable light sources, thus moving toward home-based skin cancer PDT [97].

Based on the clinical experience in dermatology reported elsewhere [23,31], and on the preclinical in vitro experience described in this article, future in vivo PDT studies can benefit from the performance of warp-knitted LEFs. On the one hand, internal natural or surgical cavities can receive knitted LEFs as an alternative to the small panels of light previously proposed for the porcine [87] or murine [22] models. On the other hand, lesions that are external or easily accessible by light may be indicated for external illumination, as has been proposed for dermatology or for small animal models [34,98]. In both cases, the good flexibility of the knitted LEF makes it a very suitable contact illumination device. Given an appropriate design and the lighting requirements specific to the selected animal model, knitted LEFs can undoubtedly be a reliable element of the abovementioned illumination strategies.

## 5. Conclusions

In this study, we presented a new optical device based on knitted light-emitting fabrics and dedicated to in vitro PDT study involving low irradiance over a long illumination period. The design of CELL-LEF is not suitable for clinical use, nor suitable for high-irradiance in vitro PDT protocols. We demonstrated that the CELL-LEF device enables simultaneous illumination of up to four multi-well plates. This device is particularly suitable for in vitro PDT studies involving a long period of illumination, as it ensures that cells maintain physiological temperature during the illumination procedure. This paper also provided a successful demonstration of CELL-LEF usability using human hepatocarcinoma and glioblastoma cell lines.

## Figures and Tables

**Figure 1 cancers-13-04109-f001:**
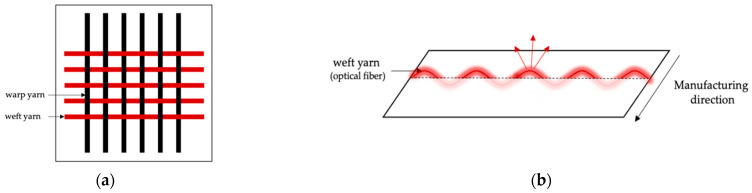
(**a**) Woven-based LEF structure. (**b**) Emission direction in a woven-based LEF.

**Figure 2 cancers-13-04109-f002:**
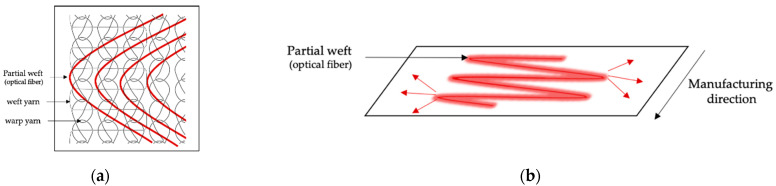
(**a**) Warp-knit-based LEF structure with OFs integrated as partial weft with the sinusoidal pattern. (**b**) Emission direction in a warp-knit-based LEF.

**Figure 3 cancers-13-04109-f003:**
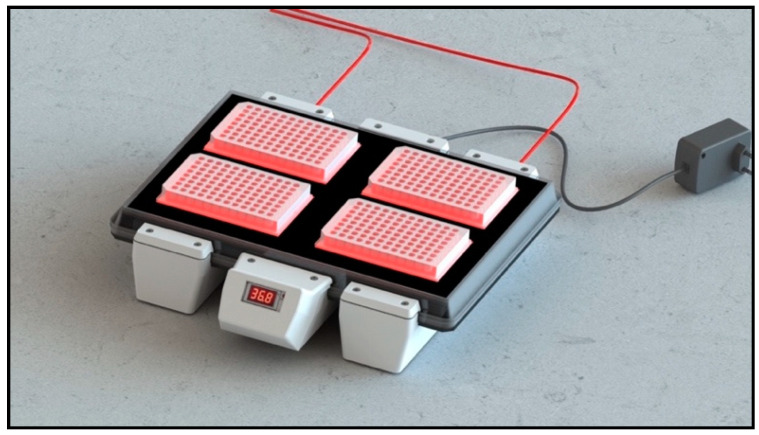
Schematic view of CELL-LEF showing the four operational areas. Each operational area has a surface of approximately 110 cm^2^ and can contain a multi-well plate.

**Figure 4 cancers-13-04109-f004:**
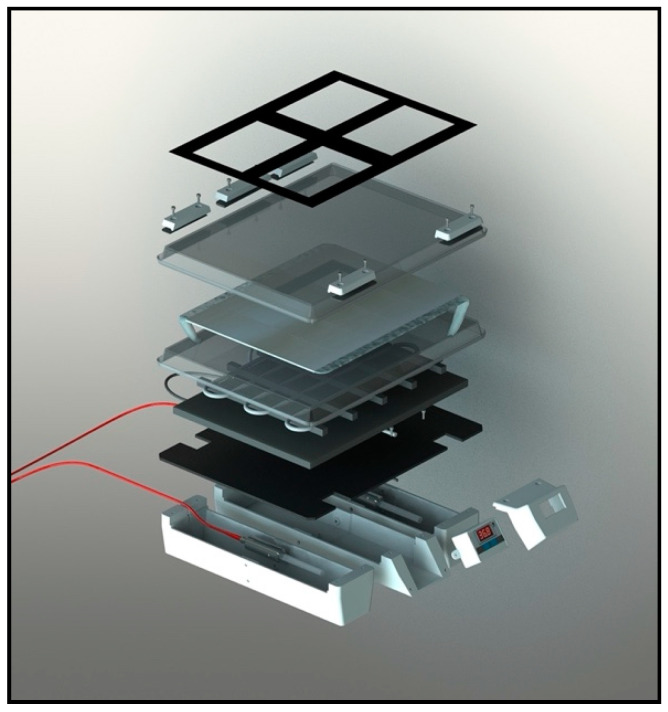
Detailed view of CELL-LEF.

**Figure 5 cancers-13-04109-f005:**
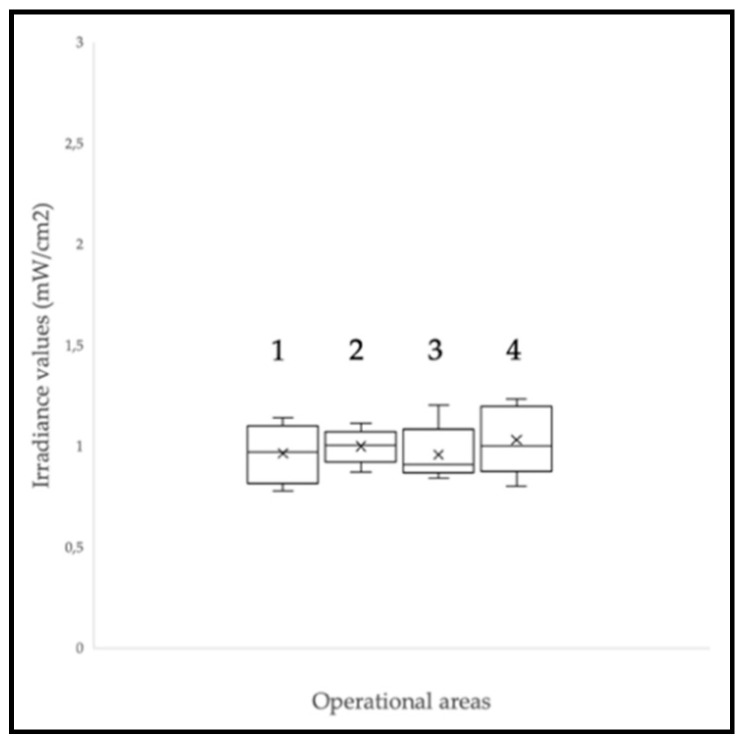
Irradiance levels at the CELL-LEF surface over the four operational areas labeled 1 to 4. The crosses correspond to the mean values, while the central horizontal bars are the median ones. The top and bottom limits of the boxes are the first and third quartiles, respectively. The ends of the whiskers represent the minimum and maximum values. As there is no point above or below the two whiskers, there is no outlier value.

**Figure 6 cancers-13-04109-f006:**
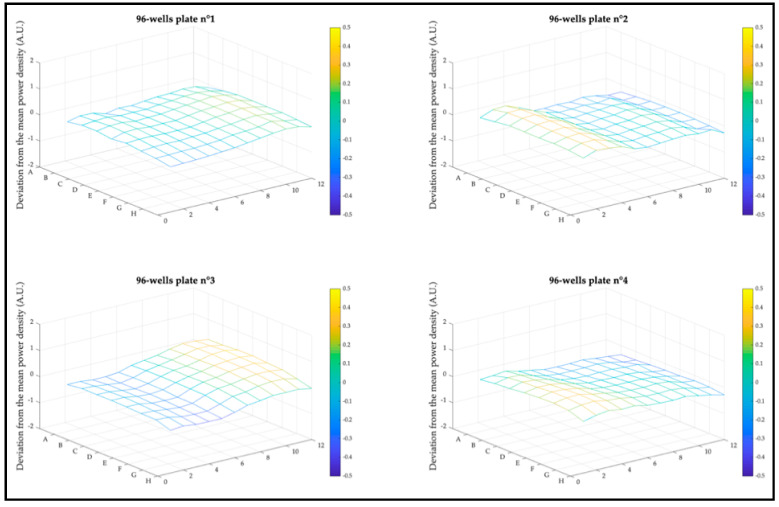
Deviation from mean irradiance distribution at well end over the four 96-well plates. The value of each well is represented by the intersection between a horizontal and a vertical line.

**Figure 7 cancers-13-04109-f007:**
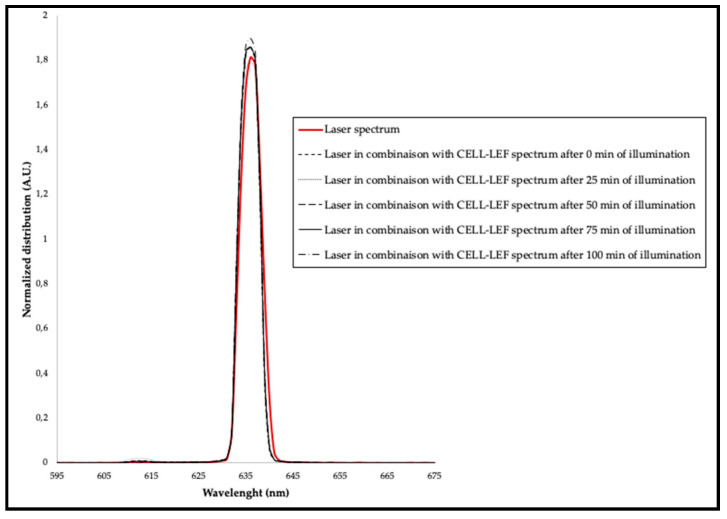
Spectral distribution of the 635 nm diode laser alone, and of the 635 nm diode laser in combination with CELL-LEF over time.

**Figure 8 cancers-13-04109-f008:**
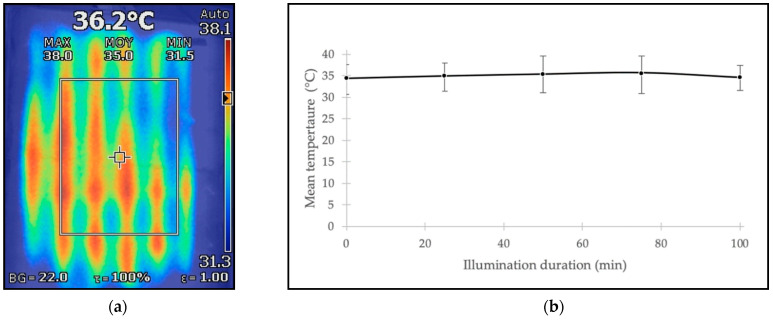
(**a**) Thermal capture of heat distribution of the CELL-LEF surface. (**b**) Mean temperature at the CELL-LEF surface every 25 min over 100 min illumination. Error bars representing the minimum and maximum values.

**Figure 9 cancers-13-04109-f009:**
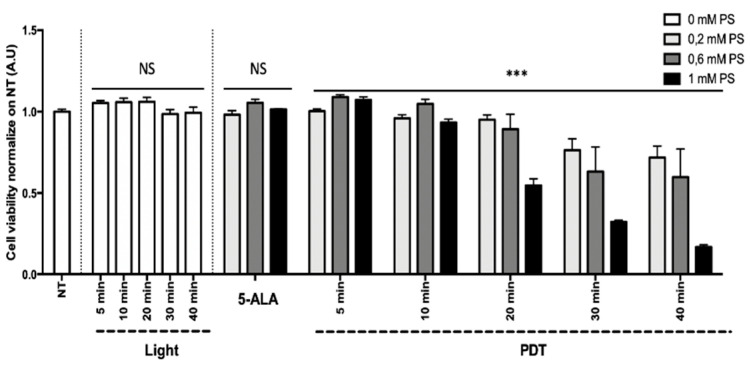
Percentage of viability at 24 h postillumination for HepG2. NT, nontreated; 5-ALA, photosensitizer only (0.2, 0.6, or 1 mM); light, illumination for 5, 10, 20, 30, and 40 min; PDT, illumination in the presence of 5-ALA for 5, 10, 20, 30, and 40 min. Results are presented as means of three independent experiments, expressed in % of the NT. Linear regression was performed, all quoted *p*-values are two-sided, with *p* ≤ 0.0001 (***) considered statistically significant, and *p* ≥ 0.05 considered not significant (NS).

**Figure 10 cancers-13-04109-f010:**
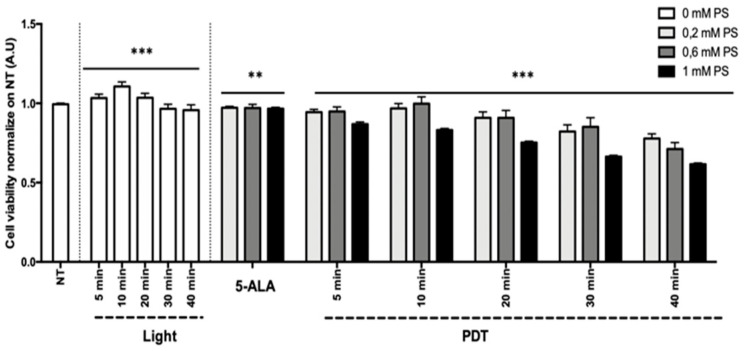
Percentage of viability at 24 h postillumination for U87. NT, nontreated; 5-ALA, photosensitizer only (0.2, 0.6,or 1 mM); light, illumination for 5, 10, 20, 30, and 40 min; PDT, illumination in the presence of 5-ALA for 5, 10, 20, 30, and 40 min. Results are presented as means of three independent experiments, expressed in % of the NT. Linear regression was performed, all quoted *p*-values are two-sided, with *p* ≤ 0.0001 (***) and *p* ≤ 0.001 (**) considered statistically significant, and *p* ≥ 0.05 considered not significant (NS).

**Figure 11 cancers-13-04109-f011:**
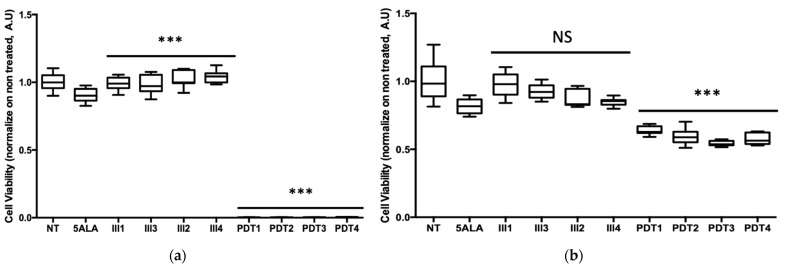
Percentage of viability at 24 h postillumination for (**a**) HepG2; (**b**) U87. NT, nontreated; 5-ALA, Photosensitizer only; Ill1, illumination in 1 fraction; Ill2, illumination in 2 fractions; Ill3, illumination in 3 fractions; Ill4, illumination in 4 fractions; PDT1, illumination in the presence of 5-ALA in 1 fraction; PDT2, illumination in the presence of 5-ALA in 2 fractions; PDT3, illumination in the presence of 5-ALA in 3 fractions; PDT4, illumination in the presence of 5-ALA in 4 fractions. Results are presented as box plots, expressed in % of the NT. ANCOVA statistical test was performed, all quoted *p*-values are two-sided, with *p* ≤ 0.0001 (***) considered statistically significant, and *p* ≥ 0.05 considered not significant (NS).

**Table 1 cancers-13-04109-t001:** Spectral radiance and risk-weighted irradiance according to the IEC 62,471 standard.

Risk Identification	Parameter	Emissions	Exposure Limits
UV	$E_{EFF}\;\left(W\cdot m^{-2}\right)$	5 × 10^−7^	5 × 10^−3^
	$E_{UV-A}\;\left(W\cdot m^{-2}\right)$	4.8 × 10^−4^	10
IR	$E_{IR}\;\left(W\cdot m^{-2}\right)$	8.7 × 10^−5^	100
Blue light	$L_{B\;}\left(W\cdot m^{-2}\cdot sr^{-1}\right)$	0.238	100
Retinal thermal	$L_{R}\;\left(W\cdot m^{-2}\cdot sr^{-1}\right)$	3111	28,000
	$L_{IR}\;\left(W\cdot m^{-2}\cdot sr^{-1}\right)$	1.61	6000

## Data Availability

The data presented in this study are available on request from the corresponding author.

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
