# Peer review of "A Warp-Knitted Light-Emitting Fabric-Based Device for In Vitro Photodynamic Therapy: Description, Characterization, and Application on Human Cancer Cell Lines"

_cancers, 2021, doi:10.3390/cancers13164109_

Round 1

Reviewer 1 Report

Please see the comments below

  1. What viability indicator was used to measure the activity using the CELL-LEF device?
  2. Are there plans to evaluate the fabric in vivo?

Reviewer 2 Report

Photodynamic therapy (PDT) is a promising strategy in biomedical application, but is limited by some factors, such as illumination source. The authors designed an illumination device, called CELL-LEF to enhance the effect of PDT. To verify the effect of CELL-LEF, the authors tested 5-ALA PDT on tumor cells.

The authors clearly introduce the device. I’m wondering if the optical design is the focus of this journal.

Regarding the effect of this device in 5-ALA PDT, I have few questions as following.

  1. Based on the cell viability shown in Figures 9 and 10, the illumination time is much longer than a typical in-vitro PDT study.
  2. The cell viability remains high (> 0.5) after 40 min illumination for U87. The effect of this PDT is not so good.

Therefore, the authors have a very special design, but it seems that the PDT effect does not get improved much.  To prove the effect of CELL-LEF, the authors can try  some control experiments with traditional light source for comparison.

Reviewer 3 Report

I don’t understand the first sentence of the summary: what is ‘setting-up? The major limitations of photodynamic therapy are [1] the need for light which can be difficult if tumor site(s) are unknown, [2] poor penetration of light into tissues if more than 5-10 mm depths are involved. This approach involves surface irradiation which might be useful for skin lesions but it is difficult to see how this would be useful for much else, e.g., esophageal, pancreatic, brain or lung cancer. It would be interesting for the authors to indicate exactly which clinical indications for PDT might utilize the approach being suggested. Skin cancer appears to be the only likely prospect. Since oxygen is required for a photodynamic effect, it is proposed that a low light dose will promote re-oxygenation during treatment. This is a reasonable approach; other protocols involve an ‘on-off’ light treatment which can also promote re-oxygenation.  

For the present studies, the device used for light delivery is also designed to provide an elevated temperature. The need for temperature control during therapy is not explained. I doubt that any clinical approach would involve anything but treatment of skin cancer and the usual approach is to cool skin, y during ALA-mediated therapy since this can be painful because of the photosensitization of nerve fibers. No persuasive rationale for any need for temperature control is proposed. It is true that biosynthesis of protoporphyrin might be reduced at lower temperatures but at the time of irradiation, this synthesis must have already occurred. ALA is not a photosensitizing agent.

The choice of tumor cells is not critical at this point. For proof of principle, any cell lines will do. But liver is usually a poor prospect since normal liver cells accumulate most photosensitizing agents at least as well as malignant cell types. Moreover, inserting light-emitting fabrics into brain seems an unlikely prospect. It would be useful to provide an explanation for the basis for the bioluminescence assay. Why not use a clonogenic assay which is much more reliable as an index of photokilling? This appears to be an assay for ATP levels which, 24 hr after irradiation, could be correlated with viability. But use of such terms as ‘cell death’ is not warranted since death is never established. The results might be expressed in terms of  ‘phototoxicity’ until loss of viability is established by clonogenic or proliferation tests. 

The difference in efficacy for HepG2 and U87 cells may derive from differences in rates of conversion of ALA to protoporphyrin or to retention of PP in the cells. At this point, this is perhaps not a major concern. Light penetration into monolayers is usually not a variable. A comparison with the Modulight system is noted. This device is used for treatment of skin cancer and has been adopted by some dermatologists. How the fabric-based device would be be superior is not explained. 

So there are aspects of this report that represent a cure looking for a disease. The only likely applications will be in the treatment of skin cancer where a need for temperature maintenance is not indicated. Some ALA-based applications involve cooling to minimize pain. It is not clear how any site other than skin could be pertinent to the proposed protocol. 

Reviewer 4 Report

This paper is on a light emitting knitted fabric-based device for PDT. I have the following remarks:

  1. The information given in the paper is not completely novel in that essentially the same technology has already, in one form or another, been presented in the literature several times (see for instance Translational Biophotonics 2(3) from August 2020, and Photodiagnosis and Photodynamic Therapy 12(1) 2015).    
  2. Such fibers were introduced into the market already by Philips in 2006 and many other companies followed. A large patent literature also exists, see for instance US4234907A or US9335457B2. A reference to some of the early work may interest the reader.
  3. References to some of the literature are missing, see for instance the work of Heinrich Walt and coworkers, who are also in the PDT field.
  4. Some of the acronyms used are not explained: Please make a list at the beginning of your publication. For instance the average reader of Cancers may simply not know what LEF (light emitting fabric?) means.
  5. Already in the abstract it is mentioned that the device leads to temperature variations between 30.7 and 38.4 degrees centigrade, this is a rather large difference. Smaller differences have been shown in the literature to influence PDT efficacy significantly (for instance in vivo see B. Henderson et. al). Please comment on this.
  6. The device described in this paper needs more than 2 Watts input to provide the 1 mW/cm2 output. This is not particularly efficient. Also, this is a VERY low fluence rate in the world of PDT. Thus the device can hardly be used conveniently for much higher fluence rates as very large and hence expensive laser systems would be needed. Please comment on this in your paper.
  7. Another laboratory that I recently visited employs another solution that is apparently often used to irradiate a large number of well plates for PDT in cell culture is to expose the wells using a frontal light distributer as produced and marketed by Rakuten Medical's company Medlight (part of Rakuten's photodynamics activity for treating cancer). These fibers are terminated by a graded index selfoc lens and have 2 perpendicular mode scramblers. Thus they provide a perfectly homogeneous circular light distribution at any distance from the fiber tip. The larger the distance from the tip the larger the circle irradiated. These fibers are also very efficient and CE marked. Losses in this fiber are very small. Thus to illuminate perfectly homogeneously a circle of 32 cm diameter which can contain a number of 96-well plates, and has a surface area of 800 cm2, if you needed only 1 mW/cm2, you could do with a laser which puts out less than 1 Watt ! This fiber has an SMA905 coupler attached to it so that it can be hooked up to most small diode lasers directly. There are no heat problems either. So why go through a complicated system you have designed? Please comment on this. 
  8. A main argument for the use of low intensities and hence longer light exposure, in the PDT of skin cancer, is less pain. You might want to add that to your arguments. The other argument (besides no oxygen limiting) is that in the case of ALA/PPIX PDT you keep on making PPIX.
  9. If I understand correctly, in your system it is the bending of the fiber that causes the light to escape from the fiber core. Now roughening of the fiber core's surface has been shown to have the same effect, and no bending is needed. Please comment on this.
  10. I understand that in your system the laser output is coupled to the input of 2200 plastic optical fibers using a metal connector. Could you please provide a clear picture of how this is accomplished.
  11. I am curious if your system provides homogeneous light output at say 400 nm, does it do the same at say 635 nm? i.e. are you independent of wavelength?
  12. You report CELL-LEF surface irradiance values between 0.78 and 1.23 mW/cm2. That is a nearly 60% difference. Is that considered acceptable? Please comment.
  13. You mention that LEDs have a larger spectral width than say a diode laser. Yes, but does that matter as most of the PS absorption spectra are also quite broad?
  14. Can your warp-knitted LEF still deliver light homogeneously when bent? This would be a major advantage when used in some medical applications!
  15. There seem to be large differences in the PDT efficiency between the two cell lines you investigated (HepG2 and U87). Maybe some of these differences come from resistance mechanisms. Resistance buildup may in some cases take time. So maybe at these very long irradiation times, PDT could become less efficient due to resistance mechanisms starting to become efficient. Could that be?  

Round 2

Reviewer 2 Report

The authors have revised the manuscript according to previously review comments. 

Reviewer 3 Report

This revised report relates to the use of a fabrication for irradiation of large surface areas in conjunction with photodynamic therapy. While there are several potential uses described, this will most likely be most useful for skin cancer. It is difficult to imagine a protocol calling for in site irradiation of brain, esophageal, GI, lung or other internal tumors that could be carried out over 80 minutes. The last sentence of the abstract is true but the relevance is unclear. ‘In vitro’ studies were never a problem with PDT. All it takes is a petri dish and a light source. The pertinent question is whether this approach will have any clinical role. 

The authors promote the use of a low light flux that can minimize hypoxia effects. While this is true, the need for excessively long irradiation periods is going to limit (as described above) clinical application. Use of an ATP assay may be adequate for an estimate of phototoxicity, but the correlation between these data and clonogenicity is unknown for many cell types. This is perhaps not a major consideration at this early point in device development. The need for temperature control is also mainly for in vitro studies since, in clinical applications, any pertinent tissue will likely be at or near 37ºC, It is true that apoptosis, the usual PDT-induced death mode, is impaired at lower temperatures. It is not clear why a heating element would be needed for, e.g., treatment of skin cancer. If there is only a heating element (line 207), how would this protect from hyperthermia (line 212)? This would require a cooling function. 

Numbers are often given to three significant figures. Since there is always a variation in biological data, a more pertinent approach would be to described results in terms of a ± b, with the standard deviation indicated. Section 2.5.3 and the legend to Figs. 9 and 10 use the term ‘viability’. If this relates to ATP data, this is not viability but toxicity. Were any clonongenic studies carried out?  

This report continues to have elements of a cure looking for a disease. Implanting light-emitting fabrics in organs has intrinsic issues, temperature control is unnecessary in clinical applications and the lengthy times involved are not going to appeal to dermatologists who already have devices for irradiation of skin tumors. 

Reviewer 4 Report

The paper is now much improved. I still have some remaining remarks:

  1. The authors agree that CELL-LEF is not suitable at high fluence rates. This is useful information for the reader and should thus of course be mentioned clearly in the paper.
  2. Please note, and include in your discussion, that clinically there are 2 main points besides the O2 consumption and the possibly rate limiting P(O2)replacement: pain which diminishes strongly with lower fluence rates, and time in the clinic (clinician's time) which is very expensive.
  3. The latter argument also holds for lab experiments, very long experiments can become quite costly because of the time, at least in some cases.
  4. I was quite surprised by the authors showing the picture from the article in a French journal entitled "la lumière qui soigne". It is well known that it is very difficult with the human eye to detect intensity changes. So I decided to contact the company where my colleague purchased his optical fibres. These fibres effectively give a - what appears to me - much better spatial irradiance than the CELL-LEF values you reported. See (https://www.medlight.com/pdf/Doc_FD1_0801E.pdf).
  5. Thank you for providing Figure 4. As I understand from the figure and the shown Gaussian distribution, the fibres in that bundle will receive very different intensities at their surface. Is this the reason why the CELL-LEF is so heterogeneous in its output. Please explain this as quantitatively as possible. Why not use the fibre I referred to above to expose the fibre bundle as this fibre does apparently not result in a Gaussian distribution, but rather in a flat topcoat distribution?
  6. There must be a typo: what is meant by: more effective WARD colon cancer destruction?
  7. As I understand from the Rakuten (Medlight) prospectus, the surface roughened fibres can be extremely homogeneous in their  "cylindrical" output along the length of the fiber. This does not agree apparently with your remarks. please discuss.
  8. You mention that in the near IR the CELL-LEF gives very different, and presumably more inhomogeneous output. With near IR used more and more for PDT this should also be clearly mentioned, if possible in a quantitative way, in the paper. 

Round 3

Reviewer 4 Report

The paper is now somewhat improved. I have the two remaining points that seem to be not clearly discussed:

  1. It is not made clear why glioblastoma cells were chosen as a model for PDT. PDT of glioblastoma has been tried and found to be rather ineffective due to the presence of so-called "guerrilla cells", i.e. small clumps of cancer cells localised up to  several cm from the main tumor bed, and hence inaccessible to the normal PDT intraoperative irradiation even at 635 nm when using ALA.
  2. The authors state that only quite low fluence rates are possible with CELL-LEF. The very long irradiation times needed for effective PDT are hence quite long and often not close to the clinical situation. How useful are then such pre-clinical studies? Please discuss this in some detail. In other words, PDT efficiency depends on light, drug concentration and location, as well as O2. If all these parameters in the preclinical study are very different from the clinical contact, how meaningful are then these preclinical studies?
